# Multi-objective Bayesian optimisation with preferences over objectives

**Majid Abdolshah, Alistair Shilton, Santu Rana, Sunil Gupta, Svetha Venkatesh**
The Applied Artificial Intelligence Institute ($A^2I^2$),
Deakin University, Australia
{majid,alistair.shilton,santu.rana,sunil.gupta,svetha.venkatesh}
@deakin.edu.au

## Abstract

We present a multi-objective Bayesian optimisation algorithm that allows the user to express preference-order constraints on the objectives of the type "objective A is more important than objective B". These preferences are defined based on the stability of the obtained solutions with respect to preferred objective functions. Rather than attempting to find a representative subset of the complete Pareto front, our algorithm selects those Pareto-optimal points that satisfy these constraints. We formulate a new acquisition function based on expected improvement in dominated hypervolume (EHI) to ensure that the subset of Pareto front satisfying the constraints is thoroughly explored. The hypervolume calculation is weighted by the probability of a point satisfying the constraints from a gradient Gaussian Process model. We demonstrate our algorithm on both synthetic and real-world problems.

## 1 Introduction

In many real world problems, practitioners are required to sequentially evaluate a noisy black-box and expensive to evaluate function $f$ with the goal of finding its optimum in some domain $\mathbb{X}$. Bayesian optimisation is a well-known algorithm for such problems. There are a variety of studies such as hyperparameter tuning [27, 13, 12], expensive multi-objective optimisation for Robotics [2, 1], and experimentation optimisation in product design such as short polymer fiber materials [16].

Multi-objective Bayesian optimisation involves at least two conflicting, black-box, and expensive to evaluate objectives to be optimised simultaneously. Multi-objective optimisation usually assumes that all objectives are *equally important*, and solutions are found by seeking the Pareto front in the objective space [4, 5, 3]. However, in most cases, users can stipulate preferences over objectives. This information will impart on the relative importance on sections of the Pareto front. Thus using this information to preferentially sample the Pareto front will boost the efficiency of the optimiser, which is particularly advantageous when the objective functions are expensive.

In this study, preferences over objectives are stipulated based on the stability of the solutions with respect to a set of objective functions. As an example, there are scenarios when investment strategists are looking for Pareto optimal investment strategies that prefer stable solutions for return (objective 1) but more diverse solutions with respect to risk (objective 2) as they can later decide their appetite for risk. As can be inferred, the stability in one objective produces more diverse solutions for the other objectives. We believe in many real-world problems our proposed method can be useful in order to reduce the cost, and improve the safety of experimental design.

Whilst multi-objective Bayesian optimisation for sample efficient discovery of Pareto front is an established research track [9, 18, 8, 15], limited work has examined the incorporation of preferences. Recently, there has been a study [18] wherein given a user specified preferred region in objective space,

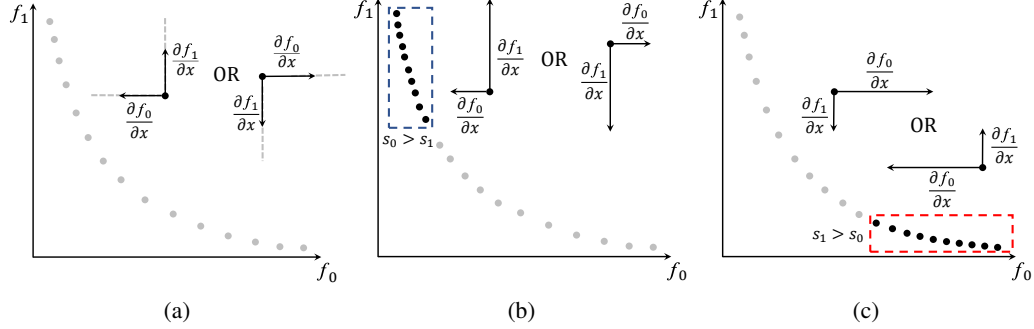

Figure 1: (a) Local Pareto optimality for 2 objective example with 1D design space. Local optimality implies $\frac{\partial f_0(x)}{\partial x}$ and $\frac{\partial f_1(x)}{\partial x}$ have opposite signs since the weighted sum of gradients of the objectives with respect to $x$ must be zero: $\mathbf{s}^{\mathrm{T}} \frac{\partial}{\partial x} \mathbf{f}(x) = 0$. In (b) we additionally require that $||\frac{\partial f_1(x)}{\partial x}|| > ||\frac{\partial f_0(x)}{\partial x}||$, so perturbation of $x$ will cause relatively more change in $f_1$ than $f_0$ - i.e. such solutions are (relatively) stable in objective $f_0$. (c) Shows the converse, namely $||\frac{\partial f_0(x)}{\partial x}|| > ||\frac{\partial f_1(x)}{\partial x}||$ favoring solutions that are (relatively) stable in objective $f_1$ and diverse in $f_0$.

the optimiser focuses its sampling to derive the Pareto front efficiently. However, such preferences are based on the assumption of having an accurate prior knowledge about objective space and the preferred region (generally a hyperbox) for Pareto front solutions. The main contribution of this study is formulating the concept of preference-order constraints and incorporating that into a multi-objective Bayesian optimisation framework to address the unavailability of prior knowledge and boosting the performance of optimisation in such scenarios.

We are formulating the preference-order constraints through ordering of derivatives and incorporating that into multi-objective optimisation using the geometry of the constraints space whilst needing no prior information about the functions. Formally, we find a representative set of Pareto-optimal solutions to the following multi-objective optimisation problem:

$$\mathbb{D}^{\star} \subset \mathbb{X}^{\star} = \underset{\mathbf{x} \in \mathbb{X}}{\operatorname{argmax}} \mathbf{f}(\mathbf{x}) \tag{1}$$

subject to *preference-order constraints* - that is, assuming $\mathbf{f} = [f_0, f_1, \ldots, f_m]$, $f_0$ is more important (in terms of stability) than $f_1$ and so on. Our algorithm aims to maximise the dominated hypervolume of the solution in a way that the solutions that meet the constraints are given more *weights*.

To formalise the concept of preference-order constraints, we first note that a point is locally Pareto optimal if any sufficiently small perturbation of a single design parameter of that point does not simultaneously increase (or decrease) all objectives. Thus, equivalently, a point is locally Pareto optimal if we can define a set of weight vectors such that, for each design parameter, the weighted sum of gradients of the objectives with respect to that design parameter is zero (see Figure 1a). Therefore, the weight vectors define the relative importance of each objective at that point. Figure 1b illustrates this concept where the blue box defines the region of stability for the function $f_0$. Since in this section the magnitude of partial derivative for $f_0$ is smaller compared to that of $f_1$, the weights required to satisfy Pareto optimality would need higher weight corresponding to the gradient of $f_0$ compared to that of $f_1$ (see Figure 1b). Conversely, in Figure 1c, the red box highlights the section of the Pareto front where solutions have high stability in $f_1$. To obtain samples from this section of the Pareto front, we need to make the weights corresponding to the gradient of $f_0$ to be smaller to that of the $f_1$.

Our solution is based on understanding the geometry of the constraints in the weight space. We show that preference order constraints gives rise to a polyhedral proper cone in this space. We show that for the pareto-optimality condition, it necessitates the gradients of the objectives at pareto-optimal points to lie in a perpendicular cone to that polyhedral. We then quantify the posterior probability that any point satisfies the preference-order constraints given a set of observations. We show how these posterior probabilities may be incorporated into the EHI acquisition function [11] to steer the Bayesian optimiser toward Pareto optimal points that satisfy the preference-order constraint and away from those that do not.

## 2 Notation

Sets are written $\mathbb{A}, \mathbb{B}, \mathbb{C}, \ldots$ where $\mathbb{R}_+$ is the positive reals, $\bar{\mathbb{R}}_+ = \mathbb{R}_+ \cup \{0\}$, $\mathbb{Z}_+ = \{1, 2, \ldots\}$, and $\mathbb{Z}_n = \{0, 1, \ldots, n-1\}$. $|\mathbb{A}|$ is the cardinality of the set $\mathbb{A}$. Tuples (ordered sets) are denoted $\mathfrak{A}, \mathfrak{B}, \mathfrak{C}, \ldots$. Distributions are denoted $\mathcal{A}, \mathcal{B}, \mathcal{C}, \ldots$. column vectors are bold lower case $\mathbf{a}, \mathbf{b}, \mathbf{c}, \ldots$. Matrices bold upper case $\mathbf{A}, \mathbf{B}, \mathbf{C}, \ldots$. Element $i$ of vector $\mathbf{a}$ is $a_i$, and element $i, j$ of matrix $\mathbf{A}$ is $A_{i,j}$ (all indexed $i, j = 0, 1, \ldots$). The transpose is denoted $\mathbf{a}^{\mathrm{T}}, \mathbf{A}^{\mathrm{T}}$. $\mathbf{I}$ is the identity matrix, $\mathbf{1}$ is a vector of 1s, $\mathbf{0}$ is a vector of 0s, and $\mathbf{e}_i$ is a vector $e_{(i)j} = \delta_{ij}$, where $\delta_{ij}$ is the Kronecker-Delta. $\nabla_{\mathbf{x}} = [\frac{\partial}{\partial x_0} \frac{\partial}{\partial x_1} \cdots \frac{\partial}{\partial x_{n-1}}]^{\mathrm{T}}$, $\mathrm{sgn}(x)$ is the sign of $x$ (where $\mathrm{sgn}(0) = 0$), and the indicator function is denoted as $\mathbb{1}(\mathtt{A})$.

## 3 Background

### 3.1 Gaussian Processes

Let $\mathbb{X} \subset \mathbb{R}^n$ be compact. A Gaussian process [23] $\mathcal{GP}(\mu, K)$ is a distribution on the function space $f : \mathbb{X} \to \mathbb{R}$ defined by mean $\mu : \mathbb{X} \to \mathbb{R}$ (assumed zero without loss of generality) and kernel (covariance) $K : \mathbb{X} \times \mathbb{X} \to \mathbb{R}$. If $f(\mathbf{x}) \sim \mathcal{GP}(0, K(\mathbf{x}, \mathbf{x}'))$ then the posterior of $f$ given $\mathbb{D} = \{(\mathbf{x}_{(j)}, y_{(j)}) \in \mathbb{R}^n \times \mathbb{R} | y_{(j)} = f(\mathbf{x}_{(j)}) + \epsilon, \epsilon \sim \mathcal{N}(0, \sigma^2), j \in \mathbb{Z}_N \}$, $f(\mathbf{x}) | \mathbb{D} \sim \mathcal{N}(\mu_{\mathbb{D}}(\mathbf{x}), \sigma_{\mathbb{D}}(\mathbf{x}, \mathbf{x}'))$, where:

$$
\begin{aligned}
\mu_{\mathbb{D}}(\mathbf{x}) &= \mathbf{k}^{\mathrm{T}}(\mathbf{x})\left(\mathbf{K} + \sigma^2 \mathbf{I}\right)^{-1}\mathbf{y} \\
\sigma_{\mathbb{D}}(\mathbf{x}, \mathbf{x}') &= K(\mathbf{x}, \mathbf{x}') - \mathbf{k}^{\mathrm{T}}(\mathbf{x})\left(\mathbf{K} + \sigma^2 \mathbf{I}\right)^{-1}\mathbf{k}(\mathbf{x}')
\end{aligned}
\tag{2}
$$

and $\mathbf{y}, \mathbf{k}(\mathbf{x}) \in \mathbb{R}^{|\mathbb{D}|}$, $\mathbf{K} \in \mathbb{R}^{|\mathbb{D}| \times |\mathbb{D}|}$, $k(\mathbf{x})_j = K(\mathbf{x}, \mathbf{x}_{(j)})$, $K_{jk} = K(\mathbf{x}_{(j)}, \mathbf{x}_{(k)})$.

Since differentiation is a linear operation, the derivative of a Gaussian process is also a Gaussian process [17, 22]. The posterior of $\nabla_{\mathbf{x}} f$ given $\mathbb{D}$ is $\nabla_{\mathbf{x}} f(\mathbf{x}) | \mathbb{D} \sim \mathcal{N}(\boldsymbol{\mu}'_{\mathbb{D}}(\mathbf{x}), \boldsymbol{\sigma}'_{\mathbb{D}}(\mathbf{x}, \mathbf{x}'))$, where:

$$
\begin{aligned}
\boldsymbol{\mu}'_{\mathbb{D}}(\mathbf{x}) &= \left(\nabla_{\mathbf{x}} \mathbf{k}^{\mathrm{T}}(\mathbf{x})\right)\left(\mathbf{K} + \sigma^2 \mathbf{I}\right)^{-1}\mathbf{y} \\
\boldsymbol{\sigma}'_{\mathbb{D}}(\mathbf{x}, \mathbf{x}') &= \nabla_{\mathbf{x}} \nabla_{\mathbf{x}'}^{\mathrm{T}} K(\mathbf{x}, \mathbf{x}') - \left(\nabla_{\mathbf{x}} \mathbf{k}^{\mathrm{T}}(\mathbf{x})\right)\left(\mathbf{K} + \sigma_i^2 \mathbf{I}\right)^{-1}\left(\nabla_{\mathbf{x}'} \mathbf{k}^{\mathrm{T}}(\mathbf{x}')\right)^{\mathrm{T}}
\end{aligned}
\tag{3}
$$

### 3.2 Multi-Objective Optimisation

A multi-objective optimisation problem has the form:

$$
\underset{\mathbf{x} \in \mathbb{X}}{\mathrm{argmax}}\, \mathbf{f}(\mathbf{x})
\tag{4}
$$

where the components of $\mathbf{f} : \mathbb{X} \subset \mathbb{R}^n \to \mathbb{Y} \subset \mathbb{R}^m$ represent the $m$ distinct objectives $f_i : \mathbb{X} \to \mathbb{R}$. $\mathbb{X}$ and $\mathbb{Y}$ are called design space and objective space, respectively. A Pareto-optimal solution is a point $\mathbf{x}^\star \in \mathbb{X}$ for which it is not possible to find another solution $\mathbf{x} \in \mathbb{X}$ such that $f_i(\mathbf{x}) > f_i(\mathbf{x}^\star)$ for all objectives $f_0, f_1, \ldots f_{m-1}$. The set of all Pareto optimal solutions is the Pareto set $\mathbb{X}^\star = \{\mathbf{x}^\star \in \mathbb{X} | \nexists \mathbf{x} \in \mathbb{X} : \mathbf{f}(\mathbf{x}) \succ \mathbf{f}(\mathbf{x}^\star)\}$ where $\mathbf{y} \succ \mathbf{y}'$ ($\mathbf{y}$ dominates $\mathbf{y}'$) means $\mathbf{y} \neq \mathbf{y}'$, $y_i \geq y_i'$ $\forall i$, and $\mathbf{y} \succeq \mathbf{y}'$ means $\mathbf{y} \succ \mathbf{y}'$ or $\mathbf{y} = \mathbf{y}'$.

Given observations $\mathbb{D} = \{(\mathbf{x}_{(j)}, \mathbf{y}_{(j)}) \in \mathbb{R}^n \times \mathbb{R}^m | \mathbf{y}_{(j)} = \mathbf{f}(\mathbf{x}_{(j)}) + \boldsymbol{\epsilon}, \epsilon_i \sim \mathcal{N}(0, \sigma_i^2)\}$ of $\mathbf{f}$ the dominant set $\mathbb{D}^* = \{(\mathbf{x}^*, \mathbf{y}^*) \in \mathbb{D} | \nexists (\mathbf{x}, \mathbf{y}) \in \mathbb{D} : \mathbf{y} \succeq \mathbf{y}^*\}$ is the most optimal subset of $\mathbb{D}$ (in the Pareto sense). The "goodness" of $\mathbb{D}$ is often measured by the dominated hypervolume ($S$-metric, [31, 10]) with respect to some reference point $\mathbf{z} \in \mathbb{R}^m$: $S(\mathbb{D}) = S(\mathbb{D}^*) = \int_{\mathbf{y} \geq \mathbf{z}} \mathbb{1}\left(\exists \mathbf{y}_{(i)} \in \mathbb{D} | \mathbf{y}_{(i)} \succeq \mathbf{y}\right) d\mathbf{y}$. Thus our aim is to find the set $\mathbb{D}$ that maximises the hypervolume. Optimised algorithms exist for calculating hypervolume [29, 25], $S(\mathbb{D})$, which is typically calculated by sorting the dominant observations along each axis in objective space to form a grid. Dominated hypervolume (with respect to $\mathbf{z}$) is then the sum of the hypervolumes of the dominated cells ($c_k$) - i.e. $S(\mathbb{D}) = \sum_k \mathrm{vol}(c_k)$.

### 3.3 Bayesian Multi-Objective Optimisation

In the multi-objective case one typically assumes that the components of $\mathbf{f}$ are draws from independent Gaussian processes, i.e. $f_i(\mathbf{x}) \sim \mathcal{GP}(0, K_{(i)}(\mathbf{x}, \mathbf{x}'))$, and $f_i$ and $f_{i'}$ are independent $\forall i \neq i'$. A

popular acquisition function for multi-objective Bayesian optimisation is expected hypervolume improvement (EHI). The EHI acquisition function is defined by:

$$a_t \left( \mathbf{x} | \, \mathbb{D} \right) = \mathbb{E}_{\mathbf{f}(\mathbf{x})|\mathbb{D}} \left[ \mathrm{S} \left( \mathbb{D} \cup \{(\mathbf{x}, \mathbf{f}\left(\mathbf{x}\right))\} \right) - \mathrm{S} \left( \mathbb{D} \right) \right] \tag{5}$$

[26, 30] and represents the expected change in the dominated hypervolume by the set of observations based on the posterior Gaussian process.

## 4    Problem Formulation

Let $\mathbf{f} : \mathbb{X} \subset \mathbb{R}^n \to \mathbb{Y} \subset \mathbb{R}^m$ be a vector of $m$ independent draws $f_i \sim \mathcal{GP}(0, K_{(i)}(\mathbf{x}, \mathbf{x}))$ from zero-mean Gaussian processes. Assume that $\mathbf{f}$ is expensive to evaluate. Our aim is to find a representative set of Pareto-optimal solutions to the following multi-objective optimisation problem:

$$\mathbb{D}^\star \subset \mathbb{X}^\star = \underset{\mathbf{x} \in \mathbb{X}_\mathfrak{I} \subset \mathbb{X}}{\operatorname{argmax}} \mathbf{f}\left(\mathbf{x}\right) \tag{6}$$

subject to *preference-order constraints*. Specifically, we want to explore only that subset of solutions $\mathbb{X}_\mathfrak{I} \subset \mathbb{X}$ that place more *importance* on one objective $f_{i_0}$ than objective $f_{i_1}$, and so on, as specified by the (ordered) preference tuple $\mathfrak{I} = (i_0, i_1, \ldots i_Q | \{i_0, i_1, \ldots\} \subset \mathbb{Z}_m, i_k \neq i_{k'} \forall k \neq k')$, where $Q \in \mathbb{Z}_m$ is the number of defined preferences over objectives.

### 4.1    Preference-Order Constraints

Let $\mathbf{x}^\star \in \mathrm{int}(\mathbb{X}) \cap \mathbb{X}^\star$ be a Pareto-optimal point in the interior of $\mathbb{X}$. Necessary (but not sufficient, local) Pareto optimality conditions require that, for all sufficiently small $\delta\mathbf{x} \in \mathbb{R}^n$, $f(\mathbf{x}^\star + \delta\mathbf{x}) \not\succ f(\mathbf{x})$, or, equivalently $\left( \delta\mathbf{x}^{\mathrm{T}} \nabla_{\mathbf{x}} \right) \mathbf{f}\left(\mathbf{x}^\star\right) \notin \mathbb{R}_+^m$. A necessary (again not sufficient) equivalent condition is that, for each axis $j \in \mathbb{Z}_n$ in design space, sufficiently small changes in $x_j$ do not cause all objectives to simultaneously increase (and/or remain unchanged) or decrease (and/or remain unchanged). Failure of this condition would indicate that simply changing design parameter $x_j$ could improve all objectives, and hence that $\mathbf{x}^\star$ was not in fact Pareto optimal. In summary, local Pareto optimality requires that $\forall j \in \mathbb{Z}_n$ there exists $\mathbf{s}_{(j)} \in \bar{\mathbb{R}}_+^m \backslash \{\mathbf{0}\}$ such that:

$$\mathbf{s}_{(j)}^{\mathrm{T}} \tfrac{\partial}{\partial x_j} \mathbf{f}\left(\mathbf{x}\right) = 0 \tag{7}$$

It is important to note that this is not the same as the optimality conditions that may be derived from linear scalarisation, as the optimality conditions that arrise from linear scalarisation additionally require that $\mathbf{s}_{(0)} = \mathbf{s}_{(1)} = \ldots = \mathbf{s}_{(n-1)}$. Moreover (7) applies to all Pareto-optimal points, whereas linear scalarisation optimisation conditions fail for Pareto points on non-convex regions [28].

**Definition 1 (Preference-Order Constraints)** *Let* $\mathfrak{I} = (i_0, i_1, \ldots i_Q | \{i_0, i_1, \ldots\} \subset \mathbb{Z}_m, i_k \neq i_{k'} \forall k \neq k')$ *be an (ordered) preference tuple. A vector* $\mathbf{x} \in \mathbb{X}$ *satisfies the associated preference-order constraint if* $\exists \mathbf{s}_{(0)}, \mathbf{s}_{(1)}, \ldots, \mathbf{s}_{(n-1)} \in \mathbb{S}_\mathfrak{I}$ *such that:*

$$\mathbf{s}_{(j)}^{\mathrm{T}} \tfrac{\partial}{\partial x_j} \mathbf{f}\left(\mathbf{x}\right) = 0 \; \forall j \in \mathbb{Z}_n$$

*where* $\mathbb{S}_\mathfrak{I} \triangleq \left\{ \mathbf{s} \in \bar{\mathbb{R}}_+^m \backslash \{\mathbf{0}\} \big| s_{i_0} \geq s_{i_1} \geq s_{i_2} \geq \ldots \right\}$. *Further we define* $\mathbb{X}_\mathfrak{I}$ *to be the set of all* $\mathbf{x} \in \mathbb{X}$ *satisfying the preference-order constraint. Equivalently:*

$$\mathbb{X}_\mathfrak{I} = \{\mathbf{x} \in \mathbb{X} | \tfrac{\partial}{\partial x_j} \mathbf{f}\left(\mathbf{x}\right) \in \mathbb{S}_\mathfrak{I}^\perp \; \forall j \in \mathbb{Z}_n \}$$

*where* $\mathbb{S}_\mathfrak{I}^\perp \triangleq \left\{ \mathbf{x} \in \mathbb{X} | \exists \mathbf{s} \in \mathbb{S}_\mathfrak{I}, \mathbf{s}^{\mathrm{T}} \mathbf{x} = 0 \right\}$.

It is noteworthy to mention that (7) and Definition 1 are the key for calculating the compliance of a recommended solution with the preference-order constraints. Having defined preference-order constraints we then calculate the posterior probability that $\mathbf{x} \in \mathbb{X}_\mathfrak{I}$, and showing how these posterior probabilities may be incorporated into the EHI acquisition function to steer the Bayesian optimiser toward Pareto optimal points that satisfy the preference-order constraint. Before proceeding, however, it is necessary to briefly consider the geometry of $\mathbb{S}_\mathfrak{I}$ and $\mathbb{S}_\mathfrak{I}^\perp$.

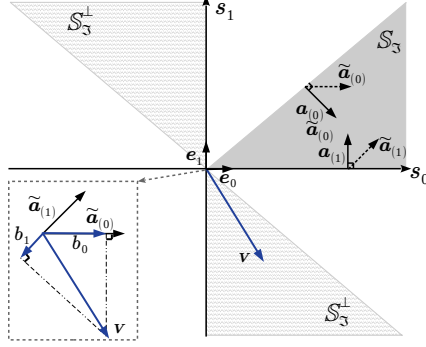

Figure 2: Illustration of $\mathbb{S}_{\mathfrak{I}}^{\perp}, \mathbb{S}_{\mathfrak{I}}$ and the vectors $\mathbf{a}_{(0)}, \mathbf{a}_{(1)}$ for a 2D case where $\mathfrak{I} = (0, 1)$, so $s_0 > s_1$, $\mathbb{S}_{\mathfrak{I}}$ is a proper cone representing the preference-order constraints; $\mathbb{S}_{\mathfrak{I}}^{\perp}$ is the union of two sub-spaces. $\mathbf{v} \in \mathbb{S}_{\mathfrak{I}}^{\perp}$ implies a solution complying with preference-order constraints. $b_0$ and $b_1$ are the projection of $\mathbf{v}$ over $\tilde{\mathbf{a}}_{(0)}$ and $\hat{\mathbf{a}}_{(1)}$. In order to satisfy $\mathbf{v} \in \mathbb{S}_{\mathfrak{I}}^{\perp}$, it is necessary that $\exists \mathbf{s} \in \mathbb{S}_{\mathfrak{I}} \ s.t. \ \mathbf{v}^T \mathbf{s} = 0$ or equivalently $\mathbf{v} = \mathbf{0}$ or $b_0 = \tilde{\mathbf{a}}_{(0)}^T \mathbf{v}$ and $b_1 = \tilde{\mathbf{a}}_{(1)}^T \mathbf{v}$ have different signs.

## 4.2 The geometry of $\mathbb{S}_{\mathfrak{I}}$ and $\mathbb{S}_{\mathfrak{I}}^{\perp}$

In the following we assume, w.l.o.g, that the preference-order constraints follows the order of indices in objective functions (reorder, otherwise), and that there is at least one constraint.

We now define the preference-order constraints by assumption $\mathfrak{I} = (0, 1, \ldots, Q | Q \in \mathbb{Z}_m \backslash \{0\})$, where $Q > 0$. This defines the sets $\mathbb{S}_{\mathfrak{I}}$ and $\mathbb{S}_{\mathfrak{I}}^{\perp}$, which in turn define the constraints that must be met by the gradients of $\mathbf{f}(\mathbf{x})$ - either $\exists \mathbf{s}_{(0)}, \mathbf{s}_{(1)}, \ldots, \mathbf{s}_{(n-1)} \in \mathbb{S}_{\mathfrak{I}}$ such that $\mathbf{s}_{(j)}^{\mathrm{T}} \frac{\partial}{\partial x_j} \mathbf{f}(\mathbf{x}) = 0 \ \forall j \in \mathbb{Z}_n$ or, equivalently $\frac{\partial}{\partial x_j} \mathbf{f}(\mathbf{x}) \in \mathbb{S}_{\mathfrak{I}}^{\perp} \ \forall j \in \mathbb{Z}_n$. Next, Theorem 1 defines the representation of $\mathbb{S}_{\mathfrak{I}}$.

**Theorem 1** *Let $\mathfrak{I} = (0, 1, \ldots, Q | Q \in \mathbb{Z}_m \backslash \{0\})$ be an (ordered) preference tuple. Define $\mathbb{S}_{\mathfrak{I}}$ as per definition 1. Then $\mathbb{S}_{\mathfrak{I}}$ is a polyhedral (finitely-generated) proper cone (excluding the origin) that may be represented using either a polyhedral representation:*

$$\mathbb{S}_{\mathfrak{I}} = \left\{ \mathbf{s} \in \mathbb{R}^m | \mathbf{a}_{(i)}^{\mathrm{T}} \mathbf{s} \geq 0 \forall i \in \mathbb{Z}_m \right\} \backslash \{\mathbf{0}\} \tag{8}$$

*or a generative representation:*

$$\mathbb{S}_{\mathfrak{I}} = \left\{ \sum_{i \in \mathbb{Z}_m} c_i \tilde{\mathbf{a}}_{(i)} \ \big| \ \mathbf{c} \in \bar{\mathbb{R}}_+^m \right\} \backslash \{\mathbf{0}\} \tag{9}$$

*where $\forall i \in \mathbb{Z}_m$:*

$$\mathbf{a}_{(i)} = \begin{cases} \frac{1}{\sqrt{2}} (\mathbf{e}_i - \mathbf{e}_{i+1}) & \text{if } i \in \mathbb{Z}_Q \\ \mathbf{e}_i & \text{otherwise} \end{cases}$$

$$\tilde{\mathbf{a}}_{(i)} = \begin{cases} \frac{1}{\sqrt{i+1}} \sum_{l \in \mathbb{Z}_{i+1}} \mathbf{e}_l & \text{if } i \in \mathbb{Z}_{Q+1} \\ \mathbf{e}_i & \text{otherwise} \end{cases}$$

*and $\mathbf{e}_0, \mathbf{e}_1, \ldots, \mathbf{e}_{m-1}$ are the Euclidean basis of $\mathbb{R}^m$.*

Proof of Theorem 1 is available in the supplementary material. To test if a point satisfies this requirement we need to understand the geometry of the set $\mathbb{S}_{\mathfrak{I}}$. The Theorem 1 shows that $\mathbb{S}_{\mathfrak{I}} \cup \{\mathbf{0}\}$ is a polyhedral (finitely generated) proper cone, represented either in terms of half-space constraints (polyhedral form) or as a positive span of extreme directions (generative representation). The geometrical intuition for this is given in Figure 2 for a simple, 2-objective case with a single preference order constraint.

---

**Algorithm 1** Test if $\mathbf{v} \in \mathbb{S}_{\mathfrak{I}}^{\perp}$.

---

**Input:** Preference tuple $\mathfrak{I}$
Test vector $\mathbf{v} \in \mathbb{R}^m$.
**Output:** $\mathbb{1}(\mathbf{v} \in \mathbb{S}_{\mathfrak{I}}^{\perp})$.
// *Calculate* $\mathbf{1}(\mathbf{v} \in \mathbb{S}_{\mathfrak{I}}^{\perp})$.
Let $b_j = \tilde{\mathbf{a}}_{(j)}^{\mathrm{T}} \mathbf{v} \; \forall j \in \mathbb{Z}_m$.
**if** $\exists i \neq k \in \mathbb{Z}_m : \mathrm{sgn}(b_i) \neq \mathrm{sgn}(b_k)$ **return**
TRUE
**elseif** $\mathbf{b} = \mathbf{0}$ **return** TRUE
**else return** FALSE.

---

**Algorithm 2** Preference-Order Constrained Bayesian Optimisation (MOBO-PC).

---

**Input:** preference-order tuple $\mathfrak{I}$.
Observations $\mathbb{D} = \{(\mathbf{x}_{(i)}, \mathbf{y}_{(i)}) \in \mathbb{X} \times \mathbb{Y}\}$.
**for** $t = 0, 1, \ldots, T-1$ **do**
  Select the test point:
    $\mathbf{x} = \underset{\mathbf{x} \in \mathbb{X}}{\mathrm{argmax}}\, a_t^{\mathrm{PEHI}}(\mathbf{x}|\mathbb{D}_t)$.
  ($a_t^{\mathrm{PEHI}}$ is evaluated using algorithm 4).
  Perform Experiment $\mathbf{y} = \mathbf{f}(\mathbf{x}) + \boldsymbol{\epsilon}$.
  Update $\mathbb{D}_{t+1} := \mathbb{D}_t \cup \{(\mathbf{x}, \mathbf{y})\}$.
**end for**

---

**Algorithm 3** Calculate $\mathrm{Pr}(\mathbf{x} \in \mathbb{X}_{\mathfrak{I}}|\mathbb{D})$.

---

**Input:** Observations $\mathbb{D} = \{(\mathbf{x}_{(i)}, \mathbf{y}_{(i)}) \in \mathbb{X} \times \mathbb{Y}\}$.
Number of Monte Carlo samples $R$.
Test vector $\mathbf{x} \in \mathbb{X}$.
**Output:** $\mathrm{Pr}(\mathbf{x} \in \mathbb{X}_{\mathfrak{I}}|\mathbb{D})$.
Let $q = 0$.
**for** $k = 0, 1, \ldots, R-1$ **do**
  //*Construct samples*
  $\mathbf{v}_{(0)}, \mathbf{v}_{(1)}, \ldots, \mathbf{v}_{(n-1)} \in \mathbb{R}^m$.
  Let $\mathbf{v}_{(j)} = \mathbf{0} \; \forall j \in \mathbb{Z}_n$.
  **for** $i = 0, 1, \ldots, m-1$ **do**
    Sample $\mathbf{u} \sim \mathcal{N}(\boldsymbol{\mu}'_{\mathbb{D}i}(\mathbf{x}), \boldsymbol{\sigma}'_{\mathbb{D}i}(\mathbf{x}, \mathbf{x}))$
    (see (3)).
    Let $[v_{(0)i}, v_{(1)i}, \ldots, v_{(n-1)i}] := \mathbf{u}^{\mathrm{T}}$.
  **end for**
  //*Test if* $\mathbf{v}_{(j)} \in \mathbb{S}_{\mathfrak{I}}^{\perp} \; \forall j \in \mathbb{Z}_n$.
  Let $q := q + \prod_{j \in \mathbb{Z}_n} \mathbb{1}(\mathbf{v}_{(j)} \in \mathbb{S}_{\mathfrak{I}}^{\perp})$ (see algorithm 1).
**end for**
Return $\frac{q}{R}$.

---

**Algorithm 4** Calculate $a_t^{\mathrm{PEHI}}(\mathbf{x}|\mathbb{D})$.

---

**Input:** Observations $\mathbb{D} = \{(\mathbf{x}_{(i)}, \mathbf{y}_{(i)}) \in \mathbb{X} \times \mathbb{Y}\}$.
Number of Monte Carlo samples $\tilde{R}$.
Test vector $\mathbf{x} \in \mathbb{X}$.
**Output:** $a_t^{\mathrm{PEHI}}(\mathbf{x}|\mathbb{D})$.
Using algorithm 3, calculate:
  $s_x = \mathrm{Pr}\left(\mathbf{x} \in \mathbb{X}_{\mathfrak{I}} \middle| \mathbb{D}\right)$
  $s_{(j)} = \mathrm{Pr}\left(\mathbf{x}_{(j)} \in \mathbb{X}_{\mathfrak{I}} \middle| \mathbb{D}\right) \; \forall (\mathbf{x}_{(j)}, \mathbf{y}_{(j)}) \in \mathbb{D}$
Let $q = 0$.
**for** $k = 0, 1, \ldots, \tilde{R} - 1$ **do**
  Sample $y_i \sim \mathcal{N}(\mu_{\mathbb{D}i}(\mathbf{x}), \sigma_{\mathbb{D}i}(\mathbf{x})))\; \forall i \in \mathbb{Z}_m$ (see (2)).
  Construct cells $c_0, c_1, \ldots$ from $\mathbb{D} \cup \{(\mathbf{x}, \mathbf{y})\}$ by sorting along each axis in objective space to form a grid.
  Calculate:
    $q = q +$
    $s_x \sum_{k : \mathbf{y} \succeq \tilde{\mathbf{y}}_{c_k}} \mathrm{vol}\left(c_k\right) \prod_{j \in \mathbb{Z}_N : \mathbf{y}_{(j)} \succeq \tilde{\mathbf{y}}_{c_k}} \left(1 - s_{(j)}\right)$
**end for**
Return $q/\tilde{R}$.

---

The subsequent corollary allows us to construct a simple algorithm (algorithm 1) to test if a vector $\mathbf{v}$ lies in the set $\mathbb{S}_{\mathfrak{I}}^{\perp}$. We will use this algorithm to test if $\frac{\partial}{\partial x_j} \mathbf{f}(\mathbf{x}) \in \mathbb{S}_{\mathfrak{I}}^{\perp} \; \forall j \in \mathbb{Z}_n$ - that is, if $\mathbf{x}$ satisfies the preference-order constraints. The proof of corollary 1 is available in the supplementary material.

**Corollary 1** *Let* $\mathfrak{I} = (0, 1, \ldots, Q | Q \in \mathbb{Z}_m \backslash \{0\})$ *be an (ordered) preference tuple. Define* $\mathbb{S}_{\mathfrak{I}}^{\perp}$ *as per definition 1. Using the notation of Theorem 1,* $\mathbf{v} \in \mathbb{S}_{\mathfrak{I}}^{\perp}$ *if and only if* $\mathbf{v} = \mathbf{0}$ *or* $\exists i \neq k \in \mathbb{Z}_m$ *such that* $\mathrm{sgn}(\tilde{\mathbf{a}}_{(i)}^{\mathrm{T}} \mathbf{v}) \neq \mathrm{sgn}(\tilde{\mathbf{a}}_{(k)}^{\mathrm{T}} \mathbf{v})$, *where* $\mathrm{sgn}(0) = 0$.

## 5 Preference Constrained Bayesian Optimisation

In this section we do two things. First, we show how the Gaussian process models of the objectives $f_i$ (and their derivatives) may be used to calculate the posterior probability that $\mathbf{x} \in \mathbb{X}_{\mathfrak{I}}$ defined by $\mathfrak{I} = (0, 1, \ldots, Q | Q \in \mathbb{Z}_m \backslash \{0\})$. Second, we show how the EHI acquisition function may be modified and calculated to incorporate these probabilities and hence only reward points that satisfy the preference-order conditions. Finally, we give our algorithm using this acquisition function.

### 5.1 Calculating Posterior Probabilities

Given that $f_i \sim \mathcal{GP}(0, K_{(i)}(\mathbf{x}, \mathbf{x}))$ are draws from independent Gaussian processes, and given observations $\mathbb{D}$, we wish to calculate the posterior probability that $\mathbf{x} \in \mathbb{X}_{\mathfrak{I}}$ -

i.e.: $\Pr\left(\mathbf{x}\in\mathbb{X}_\mathfrak{I}\mid\mathbb{D}\right)=\Pr\left(\frac{\partial}{\partial x_j}\mathbf{f}\left(\mathbf{x}\right)\in\mathbb{S}_\mathfrak{I}^\perp\;\forall j\in\mathbb{Z}_n\right)$. As $f_i\sim\mathcal{GP}(0,K_{(i)}(\mathbf{x},\mathbf{x}))$ it follows that $\nabla_\mathbf{x}f_i(\mathbf{x})\mid\mathbb{D}\sim\mathcal{N}_i\triangleq\mathcal{N}(\boldsymbol{\mu}'_{\mathbb{D}i}(\mathbf{x}),\boldsymbol{\sigma}'_{\mathbb{D}i}(\mathbf{x},\mathbf{x}'))$, as defined by (3). Hence:

$$\Pr\left(\mathbf{x}\in\mathbb{X}_\mathfrak{I}\mid\mathbb{D}\right)=\Pr\left(\begin{array}{c}\mathbf{v}_{(j)}\in\mathbb{S}_\mathfrak{I}^\perp\\\forall j\in\mathbb{Z}_n\end{array}\middle|\left[\begin{array}{c}v_{(0)i}\\v_{(1)i}\\\vdots\\v_{(n-1)i}\end{array}\right]\sim\mathcal{N}_i\;\forall i\in\mathbb{Z}_m\right)$$

where $\mathbf{v}\sim P(\nabla_\mathbf{x}\mathbf{f}\mid\mathbb{D})$. We estimate it using Monte-Carlo [6] sampling as per algorithm 3.

## 5.2 Preference-Order Constrained Bayesian Optimisation Algorithm (MOBO-PC)

Our complete Bayesian optimisation algorithm with Preference-order constraints is given in algorithm 2. The acquisition function introduced in this algorithm gives higher importance to points satisfying the preference-order constraints. Unlike standard EHI, we take expectation over both the expected experimental outcomes $f_i(\mathbf{x})\sim\mathcal{N}(\mu_{\mathbb{D}i}(\mathbf{x}),\sigma_{\mathbb{D}i}(\mathbf{x},\mathbf{x})),\forall i\in\mathbb{Z}_m$, and the probability that points $\mathbf{x}_{(i)}\in\mathbb{X}_\mathfrak{I}$ and $\mathbf{x}\in\mathbb{X}_\mathfrak{I}$ satisfy the preference-order constraints. We define our preference-based EHI acquisition function as:

$$a_t^{\mathrm{PEHI}}\left(\mathbf{x}\mid\mathbb{D}\right)=\mathbb{E}\left[\mathrm{S}_\mathfrak{I}\left(\mathbb{D}\cup\{(\mathbf{x},\mathbf{f}(\mathbf{x}))\}\right)-\mathrm{S}_\mathfrak{I}\left(\mathbb{D}\right)\mid\mathbb{D}\right]\tag{10}$$

where $S_\mathfrak{I}(\mathbb{D})$ is the hypervolume dominated by the observations $(\mathbf{x},\mathbf{y})\in\mathbb{D}$ satisfying the preference-order constraints. The calculation of $S_\mathfrak{I}(\mathbb{D})$ is illustrated in the supplementary material. The expectation of $S_\mathfrak{I}(\mathbb{D})$ given $\mathbb{D}$ is:

$$\mathbb{E}\left[S_\mathfrak{I}\left(\mathbb{D}\right)\mid\mathbb{D}\right]=\sum_k\mathrm{vol}\left(c_k\right)\Pr\left(\exists\left(\mathbf{x},\mathbf{y}\right)\in\mathbb{D}\mid\mathbf{y}\succeq\tilde{\mathbf{y}}_{c_k}\wedge\ldots\mathbf{x}\in\mathbb{X}_\mathfrak{I}\right)\ldots$$
$$=\sum_k\mathrm{vol}\left(c_k\right)\left(1-\prod_{(\mathbf{x},\mathbf{y})\in\mathbb{D}:\mathbf{y}\succeq\tilde{\mathbf{y}}_{c_k}}\left(1-\Pr\left(\mathbf{x}\in\mathbb{X}_\mathfrak{I}\mid\mathbb{D}\right)\right)\right)$$

where $\tilde{\mathbf{y}}_{c_k}$ is the dominant corner of cell $c_k$, $\mathrm{vol}(c_k)$ is the hypervolume of cell $c_k$, and the cells $c_k$ are constructed by sorting $\mathbb{D}$ along each axis in objective space. The posterior probabilities $\Pr(\mathbf{x}\in\mathbb{X}_\mathfrak{I}\mid\mathbb{D})$ are calculated using algorithm 3. It follows that:

$$a_t^{\mathrm{PEHI}}\left(\mathbf{x}\mid\mathbb{D}\right)=\Pr\left(\mathbf{x}\in\mathbb{X}_\mathfrak{I}\mid\mathbb{D}\right)\mathbb{E}\left[\sum_{k:\mathbf{y}\succeq\tilde{\mathbf{y}}_{c_k}}\mathrm{vol}\left(c_k\right)\prod_{j\in\mathbb{Z}_N:\mathbf{y}_{(j)}\succeq\tilde{\mathbf{y}}_{c_k}}\left(1-\Pr\left(\mathbf{x}_{(j)}\in\mathbb{X}_\mathfrak{I}\mid\mathbb{D}\right)\right)\Big|y_i\sim\ldots\atop\mathcal{N}\left(\mu_{\mathbb{D}i}\left(\mathbf{x}\right),\sigma_{\mathbb{D}i}\left(\mathbf{x}\right)\right)\;\forall i\in\mathbb{Z}_m\right]$$

where the cells $c_k$ are constructed using the set $\mathbb{D}\cup\{(\mathbf{x},\mathbf{y})\}$ by sorting along the axis in objective space. We estimate this acquisition function using Monte-Carlo simulation shown in algorithm 4.

## 6 Experiments

We conduct a series of experiments to test the empirical performance of our proposed method MOBO-PC and compare with other strategies. These experiments including synthetic data as well as optimizing the hyper-parameters of a feed-forward neural network. For Gaussian process, we use maximum likelihood estimation for setting hyperparameters [21].

### 6.1 Baselines

To the best of our knowledge there are no studies aiming to solve our proposed problem, however we are using PESMO, SMSego, SUR, ParEGO and EHI [9, 20, 19, 14, 7] to confirm the validity of the obtained Pareto front solutions. The obtained Pareto front must be in the ground-truth whilst also satisfying the preference-order constraints. We compare our results with MOBO-RS [18] by suitably specifying bounding boxes in the objective space that can replicate a preference-order constraint.

### 6.2 Synthetic Functions

We begin with a comparison on minimising synthetic function Schaffer function N. 1 with 2 conflicting objectives $f_0$, $f_1$ and 1-dimensional input. (see [24]). Figure 3a shows the ground-truth Pareto front

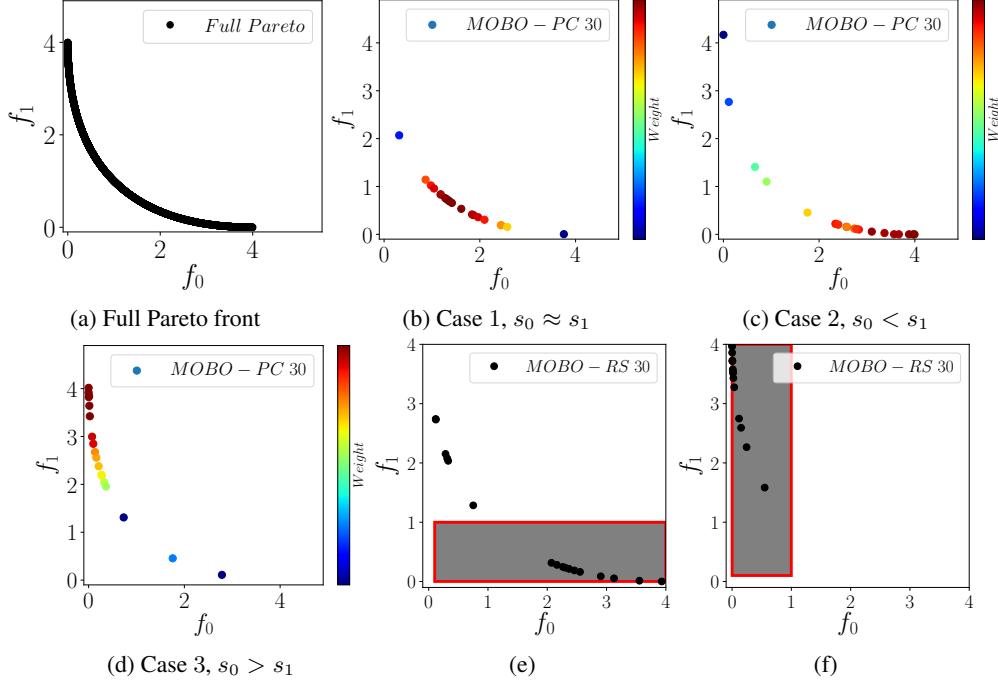

Figure 3: Finding Pareto front which comply with the preference-order constraint. Figure 3a shows the full Pareto front solution (with no preferences). Figure 3b illustrates the Pareto front by assuming stability of first objective $f_0$ is similar to second objective $f_1$. In Figure 3c, stability of $f_1$ is preferred over $f_0$. Figure 3d shows more stable results for $f_0$ than $f_1$ ($s_0 > s_1$). Figure 3e and 3f shows the results obtained by MOBO-RS and the corresponding bounding boxes. The gradient color of the Pareto front points in Figure 3b-3d indicates their degree of compliance with the constraints.

for this function. To illustrate the behavior of our method, we impose distinct preferences. Three test cases are designed to illustrate the effects of imposing preference-order constraints on the objective functions for stability. Case (1): $s_0 \approx s_1$, Case (2): $s_0 < s_1$ and Case (3): $s_0 > s_1$. For our method it is only required to define the preference-order constraints, however for MOBO-RS, additional information as a bounding box is obligatory. Figure 3b (case 1), shows the results of preference-order constraints $\mathbb{S}_{\mathfrak{I}} \triangleq \left\{ s \in \bar{\mathbb{R}}_+^m \setminus \{0\} \,\middle|\, s_0 \approx s_1 \right\}$ for our proposed method, where $s_0$ represents the importance of stability in minimising $f_0$ and $s_1$ is the importance of stability in minimising $f_1$. Due to same importance of both objectives, a balanced optimisation is expected. Higher weights are obtained for the Pareto front points in the middle region with highest stability for both objectives. Figure 3c (case 2) is based on the preference-order of $s_0 < s_1$ that implies the importance of stability in $f_1$ is more than $f_0$. The results show more stable Pareto points for $f_1$ than $f_0$. Figure 3d (case 3) shows the results of $s_0 > s_1$ preference-order constraint. As expected, we see more number of stable Pareto points for the important objective (i.e. $f_0$ in this case). We defined two bounding boxes for MOBO-RS approach which can represent the preference-order constraints in our approach (Figure 3e and 3f). There are infinite possible bounding boxes can serve as constraints on objectives in such problems, consequently, the instability of results is expected across the various definitions of bounding boxes. We believe our method can obtain more stable Pareto front solutions especially when prior information is sparse. Also, having extra information as the weight (importance) of the Pareto front points is another advantage.

Figure 4 illustrates a special test case in which $s_0 > s_1$ and $s_2 > s_1$, yet no preferences specified over $f_2$ and $f_0$ while minimising Viennet function. The proposed complex preference-order constraint does not form a proper cone as elaborated in Theorem 1. However, $s_0 > s_1$ independently constructs a proper cone, likewise for $s_2 > s_1$. Figure 4a shows the results of processing these two independent constraints separately, merging their results and finding the Pareto front. Figure 4b implies more stable solutions for $f_0$ comparing to $f_1$. Figure 4c shows the Pareto front points comply with $s_2 > s_1$.

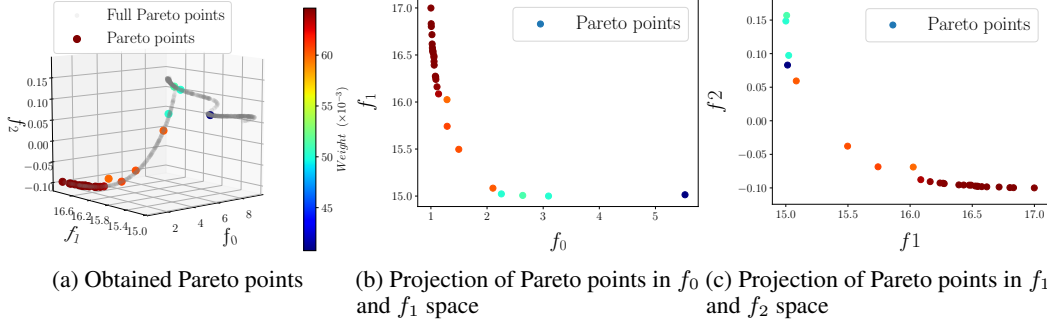

(a) Obtained Pareto points    (b) Projection of Pareto points in $f_0$    (c) Projection of Pareto points in $f_1$
                                   and $f_1$ space                            and $f_2$ space

Figure 4: Finding Pareto front points with partial constraints as specified by $s_0 > s_1$ and $s_2 > s_1$. Figure 4a shows the 3D plot of the obtained Pareto front points satisfying preference-order constraints with the color indicating the degree of compliance. Figure 4b illustrates the projection of Pareto optimal points on $f_0 \times f_1$ sub-space, and figure 4c shows the projection on $f_1 \times f_2$ sub-space.

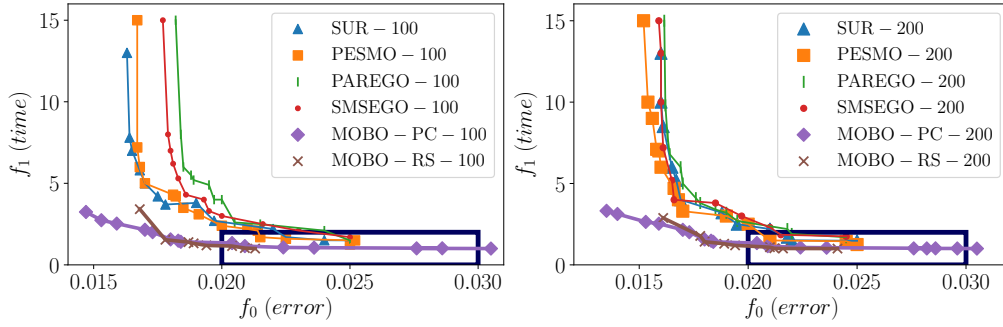

Figure 5: Average Pareto fronts obtained by proposed method in comparison to other methods. This experiment defines $s_1 > s_0$ i.e. stability of run time is more important than the error. For MOBO-RS, $[[0.02, 0], [0.03, 2]]$ is an additional information used as bounding box. The other methods do not incorporate preferences. The results are shown for 100 evaluations of the objectives (left) and 200 evaluations of the objectives (right).

### 6.3 Finding a Fast and Accurate Neural Network

Next, we train a neural network with two objectives of minimising both prediction error and prediction time, as per [9]. These are conflicting objectives because reducing the prediction error generally involves larger networks and consequently longer testing time. We are using MNIST dataset and the tuning parameters include number of hidden layers ($x_1 \in [1, 3]$), the number of hidden units per layer ($x_2 \in [50, 300]$), the learning rate ($x_3 \in (0, 0.2]$), amount of dropout ($x_4 \in [0.4, 0.8]$), and the level of $l_1$ ($x_5 \in (0, 0.1]$) and $l_2$ ($x_6 \in (0, 0.1]$) regularization. For this problem we assume stability of $f_1(time)$ in minimising procedure is more important than the $f_0(error)$. For MOBO-RS method, we selected $[[0.02, 0], [0.03, 2]]$ bounding box to represent an accurate prior knowledge (see Figure 5). The results were averaged over 5 independent runs. Figure 5 illustrates that one can simply ask for more stable solutions with respect to test time (without any prior knowledge) of a neural network while optimising the hyperparameters. As all the solutions found with MOBO-PC are in range of $(0, 5)$ test time. In addition, it seems the proposed method finds more number of Pareto front solutions in comparison with MOBO-RS.

## 7 Conclusion

In this paper we proposed a novel multi-objective Bayesian optimisation algorithm with preferences over objectives. We define objective preferences in terms of stability and formulate a common framework to focus on the sections of the Pareto front where preferred objectives are more stable, as is required. We evaluate our method on both synthetic and real-world problems and show that the obtained Pareto fronts comply with the preference-order constraints.

## Acknowledgments

This research was partially funded by Australian Government through the Australian Research Council (ARC). Prof Venkatesh is the recipient of an ARC Australian Laureate Fellowship (FL170100006).

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
