[Supplementary Material · neurips_2019_sup.pdf]

# Multi-objective Bayesian optimisation with preferences over objectives (Supplementary Materials)

**Majid Abdolshah, Alistair Shilton, Santu Rana, Sunil Gupa, Svetha Venkatesh**
The Applied Artificial Intelligence Institute $(\text{A}^2\text{I}^2)$,
Deakin University, Australia
{majid,alistair.shilton,santu.rana,sunil.gupta,svetha.venkatesh}
@deakin.edu.au

## 1   Method of Evaluation

For a more precise evaluation of our proposed method, we can define a measurement by checking how many of the Pareto front solutions satisfy the preference-order constraints. Based on Algorithm 3, we can calculate the percentage of solutions that satisfy the preference-order constraints by using the gradients of the actual synthetic functions at iteration $t$. In real-world problems, we may use the gradients of the trained Gaussian Process to evaluate the compliance of Pareto front with preference-order constraints.

It is noteworthy to mention that our function evaluations are expensive, and hence, throwing away evaluations during post-processing is undesirable. Our approach, in contrast, samples such that most of the function evaluations would have desirable characteristics, and hence, would be efficient. Considering Figure 1, given a preference-order constraint as "stability of $f_0$ being more important than $f_1$" in Schaffer function N. 1, i.e. $||\frac{\partial f_0}{\partial x}|| \leq ||\frac{\partial f_1}{\partial x}||$, Figure 1 (a) illustrates the Pareto front obtained by a plain multi-objective optimisation (with no constraints). After the Pareto solutions are found (in 20 iterations), using the derivatives of the trained Gaussian Processes (actual objective functions are black-box), we can post process the obtained Pareto front based on the stability of solutions. Figure 1 (a) shows that only $\frac{6}{18}$ of these solutions have actually met the preference-order constraints. Whereas Figure 1 (b) shows that $\frac{16}{16}$ of the obtained Pareto front solutions by MOBO-PC (in the same 20 iterations) have met the preference-order constraints.

In general, our experimental results show 98.8% of solutions found for Schaffer function N. 1 after 20 iterations comply with constraints. As for Poloni's two objective function, 86.3% of the solutions follow the constraints after 200 iterations and finally for Viennet 3D function, this number is 82.5%.

Given that the prior knowledge is not provided in [2], the obtained results for their method with same experimental design and same number of iterations are 47.2% for Schaffer function N. 1, 29.6% for Poloni's two objective function and 19.3% for Viennet 3D function respectively. This gap explains the importance of the prior knowledge about hyperboxes for their method. The reported numbers are averaged over 10 independent runs. Table 1 summarises the obtained results.

(a) MOBO-PC               (b) Naive Approach

Figure 1: (a) Illustrates a naive approach of post-processing in which almost 50% of the solutions are not complying with preference-order constraints and must be disregarded. (b) Shows an obtained Pareto front by MOBO-PC. All of the obtained solutions are complying with the preference-order constraints based on the derivative of the trained Gaussian Processes.

Table 1: Percentage of the Pareto front solutions complying with preference-order constraints in different synthetic functions.

|  | Schaffer function N. 1 | Poloni | Viennet 3D |
|---|---|---|---|
| **MOBO-PC** | 98.8% | 86.3% | 82.5% |
| **MOBO-RS** | 47.2% | 29.6% | 19.3% |

## 2 Proofs

**Theorem 1** *Let $\mathfrak{I} = (0, 1, \ldots, Q | Q \in \mathbb{Z}_m \backslash \{0\})$ be an (ordered) preference tuple. Define $\mathbb{S}_{\mathfrak{I}}$ as per definition 1. Then $\mathbb{S}_{\mathfrak{I}}$ is a polyhedral (finitely-generated) proper cone (excluding the origin) that may be represented using either a polyhedral representation:*

$$\mathbb{S}_{\mathfrak{I}} = \left\{ \mathbf{s} \in \mathbb{R}^m \, | \, \mathbf{a}_{(i)}^{\mathrm{T}} \mathbf{s} \geq 0 \forall i \in \mathbb{Z}_m \right\} \backslash \{\mathbf{0}\} \tag{1}$$

*or a generative representation:*

$$\mathbb{S}_{\mathfrak{I}} = \left\{ \sum_{i \in \mathbb{Z}_m} c_i \tilde{\mathbf{a}}_{(i)} \, \big| \, \mathbf{c} \in \bar{\mathbb{R}}_+^m \right\} \backslash \{\mathbf{0}\} \tag{2}$$

*where $\forall i \in \mathbb{Z}_m$:*

$$\mathbf{a}_{(i)} = \begin{cases} \frac{1}{\sqrt{2}} (\mathbf{e}_i - \mathbf{e}_{i+1}) & \text{if } i \in \mathbb{Z}_Q \\ \mathbf{e}_i & \text{otherwise} \end{cases}$$

$$\tilde{\mathbf{a}}_{(i)} = \begin{cases} \frac{1}{\sqrt{i+1}} \sum_{l \in \mathbb{Z}_{i+1}} \mathbf{e}_l & \text{if } i \in \mathbb{Z}_{Q+1} \\ \mathbf{e}_i & \text{otherwise} \end{cases}$$

*and $\mathbf{e}_0, \mathbf{e}_1, \ldots, \mathbf{e}_{m-1}$ are the Euclidean basis of $\mathbb{R}^m$.*

**Proof:** The polyhedral representation follows directly from consideration of the constraints $s_i \geq 0$, from which we derive the constraints $\mathbf{a}_{(i)}^{\mathrm{T}} \mathbf{s} = s_i \geq 0 \; \forall i \notin \mathbb{Z}_Q$; and the constraints $s_k \geq s_{k+1}$ $\forall k \in \mathbb{Z}_Q$, from which we derive the constraints $\mathbf{a}_{(k)}^{\mathrm{T}} \mathbf{s} = s_k - s_{k+1} \geq 0 \; \forall k = 0, 1, \ldots, Q-1$.

Moreover $\mathbb{S}_\mathfrak{I} \cup \{\mathbf{0}\} \subset \bar{\mathbb{R}}_+^m$ is constructed by restricting $\bar{\mathbb{R}}_+^m$ using half-space constraints, so $\mathbb{S}_\mathfrak{I} \cup \{\mathbf{0}\}$ is a proper cone.

The generative representation follows from the fact that a proper conic polyhedra is positively spanned by it's extreme directions ($\tilde{\mathbf{a}}_{(i)}$) - i.e. the intersections of the hyperplanes $\mathbf{a}_{(i)}^T \mathbf{s} = 0$. So $\forall i \in \mathbb{Z}_m$, $\tilde{\mathbf{a}}_{(i)}$ must satisfy:

$$\tilde{\mathbf{a}}_{(i)}^T \mathbf{a}_{(j)} = 0 \ \forall j \neq i \tag{3}$$

There are six cases that are possible combinations of $i$ and $j$. We show how (3) holds in all the conditions.

1. $i \in \{0, 1, \ldots, Q-1\}$ and $j \in \{Q, Q+1, \ldots, m-1\}$: considering theorem 1, $\tilde{\mathbf{a}}_{(i)}^T \mathbf{a}_{(j)} = \tilde{\mathbf{a}}_{(i)}^T \mathbf{e}_j = 0$. Therefore $\tilde{\mathbf{a}}_{(i)j} = 0$. Which implies $\tilde{\mathbf{a}}_{(i)Q} = \tilde{\mathbf{a}}_{(i)Q+1} = \ldots = \tilde{\mathbf{a}}_{(i)m-1} = 0$.

2. $i \in \{0, 1, \ldots, Q-1\}$ and $j \in \{0, 1, \ldots, Q-1\}\backslash\{i\}$: based on theorem 1, $\tilde{\mathbf{a}}_{(i)}^T \mathbf{a}_{(j)} = \tilde{\mathbf{a}}_{(i)}^T \frac{1}{\sqrt{2}}(\mathbf{e}_j - \mathbf{e}_{j+1}) = \frac{1}{\sqrt{2}}(\tilde{\mathbf{a}}_{(i)j} - \tilde{\mathbf{a}}_{(i)j+1}) = 0$. Therefore $\tilde{\mathbf{a}}_{(i)j} = \tilde{\mathbf{a}}_{(i)j+1}$. Hence $\tilde{\mathbf{a}}_{(i)0} = \tilde{\mathbf{a}}_{(i)1} = \ldots = \tilde{\mathbf{a}}_{(i)i}$ and $\tilde{\mathbf{a}}_{(i)i+1} = \tilde{\mathbf{a}}_{(i)i+2} \ldots = \tilde{\mathbf{a}}_{(i)Q} = 0$. Which results in $\tilde{\mathbf{a}}_{(i)} = \frac{1}{\sqrt{i+1}} \sum_{l \in \mathbb{Z}_{i+1}} \mathbf{e}_l$.

3. $i \in \{Q+1, Q+2, \ldots, m-1\}$ and $j \in \{Q, Q+1, \ldots, m-1\}\backslash\{i\}$: likewise, $\tilde{\mathbf{a}}_{(i)}^T \mathbf{a}_{(j)} = \tilde{\mathbf{a}}_{(i)}^T \mathbf{e}_j = \tilde{\mathbf{a}}_{(i)j} = 0$. Hence $\tilde{\mathbf{a}}_{(i)Q} = \tilde{\mathbf{a}}_{(i)Q+1} = \ldots = \tilde{\mathbf{a}}_{(i)m-1} = 0$, excluding $\tilde{\mathbf{a}}_{(i)i}$, where $\tilde{\mathbf{a}}_{(i)i} \neq 0$ since $j \neq i$.

4. $i \in \{Q+1, Q+2, \ldots, m-1\}$ and $j \in \{0, 1, \ldots, Q-1\}\backslash\{i\}$: we know $\tilde{\mathbf{a}}_{(i)}^T \mathbf{a}_{(j)} = \tilde{\mathbf{a}}_{(i)}^T \frac{1}{\sqrt{2}}(\mathbf{e}_j - \mathbf{e}_{j+1}) = 0$. Likewise $\tilde{\mathbf{a}}_{(i)0} = \tilde{\mathbf{a}}_{(i)1} = \ldots = \tilde{\mathbf{a}}_{(i)Q} = 0$.

5. $i = Q$ and $j \in \{Q+1, \ldots, m-1\}$: Since $i \neq j$, then $j \in \{Q+1, Q+2, \ldots, m-1\}$. Hence $\tilde{\mathbf{a}}_{(i)}^T \mathbf{a}_{(j)} = \tilde{\mathbf{a}}_{(i)}^T \mathbf{e}_j = \tilde{\mathbf{a}}_{(i)j} = 0$. Therefore $\tilde{\mathbf{a}}_{(i)Q+1} = \ldots = \tilde{\mathbf{a}}_{(i)m-1} = 0$.

6. $i = Q$ and $j \in \{0, 1, \ldots, Q-1\}$: By defining $i = Q$, we know that $i \neq j$, hence $\tilde{\mathbf{a}}_{(i)}^T \mathbf{a}_{(j)} = \tilde{\mathbf{a}}_{(i)}^T \frac{1}{\sqrt{2}}(\mathbf{e}_j - \mathbf{e}_{j+1}) = \frac{1}{\sqrt{2}}(\tilde{\mathbf{a}}_{(i)j} - \tilde{\mathbf{a}}_{(i)j+1}) = 0$, which implies $\tilde{\mathbf{a}}_{(i)0} = \tilde{\mathbf{a}}_{(i)1} = \ldots = \tilde{\mathbf{a}}_{(i)Q}$, or likewise $\tilde{\mathbf{a}}_{(i)} = \frac{1}{\sqrt{Q+1}} \sum_{l \in \mathbb{Z}_{Q+1}} \mathbf{e}_l$.

$\square$

**Corollary 1** *Let $\mathfrak{I} = (0, 1, \ldots, Q | Q \in \mathbb{Z}_m \backslash \{0\})$ be an (ordered) preference tuple. Define $\mathbb{S}_\mathfrak{I}^\perp$ as per definition 1. Using the notation of theorem 1, $\mathbf{v} \in \mathbb{S}_\mathfrak{I}^\perp$ if and only if $\mathbf{v} = \mathbf{0}$ or $\exists i \neq k \in \mathbb{Z}_m$ such that $\text{sgn}(\tilde{\mathbf{a}}_{(i)}^T \mathbf{v}) \neq \text{sgn}(\tilde{\mathbf{a}}_{(k)}^T \mathbf{v})$, where $\text{sgn}(0) = 0$.*

**Proof:** By definition of $\mathbb{S}_\mathfrak{I}^\perp$, $\mathbf{v} \in \mathbb{S}_\mathfrak{I}^\perp$ if $\exists \mathbf{s} \in \mathbb{S}_\mathfrak{I}$ such that $\mathbf{s}^T \mathbf{v} = 0$. This is trivially true of $\mathbf{v} = \mathbf{0}$. Otherwise, using the generative representation of $\mathbb{S}_\mathfrak{I}$, there must exist $\mathbf{c} \in \bar{\mathbb{R}}_+^m \backslash \{\mathbf{0}\}$ such that $\mathbf{s} = \sum_i c_i \tilde{\mathbf{a}}_{(i)}$. Hence $\mathbf{v} \in \mathbb{S}_\mathfrak{I}^\perp \backslash \{\mathbf{0}\}$ only if there exists $\mathbf{c} \in \bar{\mathbb{R}}_+^m \backslash \{\mathbf{0}\}$ such that $\mathbf{s}^T \mathbf{v} = \sum_i c_i (\mathbf{a}_{(i)}^T \mathbf{v}) = 0$ which, as $\mathbf{c} \neq \mathbf{0}$, is only possible if $\exists i, k \in \mathbb{Z}_m$ such that $\text{sgn}(\tilde{\mathbf{a}}_{(i)}^T \mathbf{v}) \neq \text{sgn}(\tilde{\mathbf{a}}_{(k)}^T \mathbf{v})$. $\square$

## 3 Experiments

### 3.1 Poloni's two objective function

The results of our algorithm on Poloni's two objective function [1]. Figure 3b shows more stable results for $f_0$ than $f_1$. Likewise, in figure 3c stability of solutions in $f_1$ is favored over $f_0$.

### 3.2 Progress of MOBO-PC

The calculation of hypervolume in MOBO-PC relies on both values of the weights for the Pareto front points and also the improved volume. That will result in favoring solutions complying with the constraints and assign them higher weights comparing to other Pareto front solutions. But if the amount of volume to be improved is insignificant, the acquisition function favors solutions with

(a) Iteration number 10  (b) Iteration number 20  (c) Iteration number 30

(d) Iteration number 40  (e) Iteration number 50  (f) Iteration number 60

Figure 2: Illustration of the progress of MOBO-PC on Schaffer function N. 1.

(a) Full Pareto front  (b) $s_0 > s_1$ preference-order con-straint  (c) $s_1 > s_0$ preference-order con-straint

Figure 3: Obtained solutions for Poloni's two objective function. 3a shows the full Pareto front. 3b illustrates the obtained solutions with $s_0 > s_1$ preference-order constraint on stability. And 3c shows the results of $s_1 > s_0$ or more stable solutions for $f_1$.

lesser compliance with constraints but more possibility to increase the amount of volume -i.e. the region with most compliance with constraints is well explored and occupied with many Pareto front solutions, so the amount of hypervolume improvement drops due to the small amount of improvement in the volume despite of higher weights for the solutions in that region. Hence, the algorithm will then look for more diverse solutions that can increase the amount of hypervolume, that is the solutions which are more diverse and less stable. Figure 2 illustrates how acquisition function favors the points with higher weights at the first, and then lean towards the more unexplored regions with higher amount of hypervolume improvement.

## 4 Simple Crash Study

### 4.0.1 Summary

This problem concerns a collision in which a simplified vehicle moving at constant velocity crashes into a pole. The input parameters vary the strength of the bumper and hood of the car. During the crash, the front portion of the car deforms. The design goal is to maximise the *crashworthiness* of the vehicle. If the car is too rigid, the passenger experiences injury due to excessive forces during the

(a) Full Pareto front  (b) $s_0 > s_1$ preference-order con-(c) $s_1 > s_0$ preference-order con-
straint                 straint

Figure 4: Obtained solutions for Simple Crash Study.

impact. If the car is not rigid enough, the passenger may be crushed as the front of the car intrudes into the passenger space. This dataset is available in https://bit.ly/2FWNXQS.

### 4.0.2 Input Variables

- tbumper is the mass of the front bumper bar. Range of tbumper is between $1$ and $5$.
- thood is the mass of the front, hood and underside of the bonnet. Range of thood is between $1$ and $5$.

### 4.0.3 Output Responses

- Intrusion is the intrusion of the car frame into the passenger space. This is computed from the change in the separation of two points, one on the front of the car (Node #167) and one of the roof (Node #432). Lower intrusion is better. Increasing the mass of the hood and bumper will reduce the intrusion.
- HIC is the head injury coefficient. This is computed from the maximum deceleration during the collision. Lower HIC is better. Increasing the mass of the hood and bumper will increase the HIC.
- Mass is the combined mass of the front structural components. Lower mass is better.

In our experiment, we are using HIC as $f_0$ and Mass as $f_1$ to be minimised simultaneously.

Figure 4 shows the obtained results for this problem. Figure 4a illustrates full Pareto front, as 4b and 4c demonstrates our obtained results based on the defined preference-order constraints. Figure 4b shows that the Pareto front points are more stable in $f_0$ (HIC) than $f_1$ (Mass). As for figure 4c Pareto front solutions are in favor of stability for $f_1$ and more diversity for $f_0$.