[Reviews · NeurIPS 2019]

Reviewer 1



Originality: The work is original in its setup of preference order in multi-objective bayesian optimisation. It extends hypervolume based acquisition function for BO with an algorithm that tests for satisfiability of preference order in a sample. Quality and clarity: The work done is complete in its motivation, formulation, aproach and experimentation. It is clearly presented. Significance: To my understanding, the use of algorithm (1) in checking if v \in \matbb{S}_{\mathcal{J}} is the only step where preference order plays a role. However, I would be interested to know how this compares to a trivial approach where the preference order sorting is carried out as a post processing step for filtering a pareto front obtained without consideration of any preference order.

Reviewer 2



There are some related works on preference-based or interactive multiobjective optimization. The novelty of the work is not very high. From my point of view, the main algorithm in Section 4 is not clearly presented. To access the preference-based approaches, the measurements are quite important. However, this issue is very subjective. The paper did not discuss this issue. Therefore, it is hard to judge whether the found solutions are good or not.

Reviewer 3



Summary: This paper proposes a method for multi-objective Bayesian optimization when a user has given “preference order constraints”, i.e. preferences about the importance of different objectives. For example, a user might specify that he or she wants to determine where, along the pareto front, a given objective varies significantly with respect to other objectives (which the authors term “diversity”) or when the objective is static with respect to other objectives (which they term “stability”). The authors give algorithms for this setting and show empirical results on synthetic functions and on a model search task. Comments: > My main criticism of this paper is that I am not convinced about the motivation for, and uses cases of, the described task of finding regions of the pareto front where an objective is “diverse” or “stable” as they are defined in the paper. There are two potential examples given in the introduction, but these are brief and unconvincing (another comment on these below). A real experiment is shown on a neural network model search task, but it is unclear how the method, when applied here, provides real benefits over other multi-objective optimization methods. More written discussion on the benefits and application of this method (for example in the model search task) could help alleviate this issue. > The three examples given in the introduction are: - A case where both objectives have constraints (precision>=0.8, recall>=0.7). - A case where we want diverse objective values along the pareto front. - A case where we want regions of the pareto front where a large change in one objective is required to obtain a small improvement in the other objective. Intuitively, these all seem to constrain the pareto front or prioritize regions of the pareto front over others. The abstract describes these as “constraints on the objectives of the type ‘objective A is more important than objective B’”. I feel that the introduction does not clearly describe how the description in the abstract aligns with the three examples given in the introduction. Is the argument that diversity/stability is a property that directly corresponds to the importance of an objective? It would be great if you could provide better clarity on this definition. > The dominated hypervolume is defined in section 2.2. It would be valuable to give some intuition about this quantity, in addition to the definition, in order to provide some clarity on how it will be used. ---------- Update after author response ---------- I want to thank the authors for their response. I believe the authors description of a couple real world examples are nice, but do not shed much light on the motivations for this method beyond the original submission. While appreciated, I will not change my score.

[Author Response · NeurIPS 2019]

We thank the reviewers for their insightful comments. In the following we only address the major issues. The manuscript will be updated accordingly to reflect the clarifications made here.

**Reviewer #1 (I) ["Comparison to naive post processing"]:** We recall that our function evaluations are expensive, and hence, throwing away evaluations during post-processing is undesirable. Our approach, in contrast, samples such that most of the function evaluations would have desirable characteristics, and hence, would be efficient. Consider the plots in Fig 1, given a preference-order constraint as "stability of $f_0$ being more important than $f_1$" in Schaffer function N. 1, i.e. $||\frac{\partial f_0}{\partial x}|| \leq ||\frac{\partial f_1}{\partial x}||$, Fig 1(left) illustrates the Pareto front obtained by a plain multi-objective optimisation (with no constraints). After the Pareto solutions are found (in 20 iterations), using the derivatives of the trained Gaussian Processes (actual objective functions are black-box), we can post process the obtained Pareto front based on the stability of solutions (lines $46 - 62$ of the paper). Fig 1(left) shows that only $\frac{6}{18}$ of these solutions have actually met the preference-order constraints. Whereas Fig 1 (right) shows that $\frac{16}{16}$ of the obtained Pareto front solutions by MOBO-PC (in the same 20 iterations) have met the preference-order constraints.

**Reviewer #1 (II), Reviewer #2 (I), Reviewer #3 (I) ["Background and related works"]:** Including preferences over objectives in MOO problems for expensive functions dates back to Hakanen et al. [1]. The authors proposed an interactive version of the ParEGO algorithm for identifying "most preferred solutions". At each interaction, the decision maker is shown a subset of non-dominated solutions and she is assumed to provide her preferences in the form of preferred ranges for each objective. Internally, the algorithm samples reference points within the **hyperbox** defined those preferred ranges. This study required both **interaction with user** at each iteration and also **prior knowledge about these hyperboxes**. Recently, Paria et al. [14] (line 330 of paper) introduced a new method to handle such constraints.

Figure 1: Comparison between a naive post processing approach (left) and MOBO-PC (right).

However this method still requires prior knowledge about the hyperboxes of the form $[[y_1, ..., y_m], [y'_1, ..., y'_m]]$ as exact location of the hyperbox in the objective function space ($\mathbb{R}^m$). We were motivated to remove the need for such complex prior information. Our proposed method achieves this as it only needs information of kind "objective $A$ is more important than objective $B$", and nothing else. We also note that **evolutionary methods** are not discussed in this paper as they require many evaluations, and hence are not suitable for inexpensive functions.

**Reviewer #2 (II) ["Measurement of performance"]:** We appreciate this question and agree that our current method of comparison through plots is subjective. However, we can define a measurement by checking how many of the Pareto front solutions satisfy the preference-order constraints. Based on **Algorithm 3** (line 202 of paper), we can calculate **the percentage of solutions that satisfy the preference-order constraints** by using the gradients of the actual functions at iteration $t$. For example, in the case of Fig 1, all of the obtained solutions are complying with stability preference-order constraints. Our experimental results show $98.8\%$ of solutions found for Schaffer function N. 1 after 20 iterations comply with constraints. As for Poloni's two objective function, $86.3\%$ of the solutions follow the constraints after 200 iterations and finally for Viennet 3D function, this number is $82.5\%$. Given that the prior knowledge is not provided in [14] (line 330 of paper), the obtained results for their method with same experimental design and same number of iterations are $47.2\%$ for Schaffer function N. 1, $29.6\%$ for Poloni's two objective function and $19.3\%$ for Viennet 3D function respectively. This gap explains **the importance of the prior knowledge** about hyperboxes for their method. The reported numbers are averaged over 10 independent runs. We will include the comprehensive results based on the iteration number in the final version of the paper.

**Reviewer #3 (II) ["Usefulness and real-world example"]:** We will use two real-world examples on **stability and diversity** to better illustrate the usefulness of MOBO-PC. **(a) Stability:** According to Chow et al. [2] a drug must be tested for **stability** before it can be released for human use. Testing the drugs on humans is a costly and potentially dangerous procedure. There are some vital signs routinely monitored (e.g. heart rate) in the testing procedure and the dosage of the drugs to be tested must be selected in a way that the practitioner can confidently confirm the **positive effects of the drug (objective 1)**, yet make sure the **vital signs such as heart rate (objective 2)** remain stable. Considering these two objectives, finding stable solutions with respect to heart rate is essential. **(b) Diversity:** There are scenarios when diversity is crucial, e.g. the investment strategists generally looking for Pareto optimal investment strategies that prefer diversity in **risk (objective 1)** over **return (objective 2)** as they can later decide their appetite for risk. **(c) Neural networks:** As in neural network example (line 277 of paper), the goal is to illustrate that one can simply ask for more stable solutions with respect to training time of a neural network while optimising the hyperparameters. As all the solutions found with MOBO-PC are in range of $(0, 5)$ training time (unlike the other methods).

[1] On using decision maker preferences with ParEGO, *International Conference on Evolutionary Multi-Criterion Optimization, 282–297, 2017*

[2] Statistical Designs for Pharmaceutical/Clinical Development *Drug Designing, 2169-0138, 2014*


[Meta-Review · NeurIPS 2019]

All the reviewers agree that the paper presents an interesting result and is nicely written. Please incorporate reviewers' feedback. Congratulations on a nice result.